# Binding mechanism of the matrix domain of HIV-1 gag on lipid membranes

Viviana Monje-Galvan, Gregory A Voth*

Department of Chemistry, Chicago Center for Theoretical Chemistry, Institute for Biophysical Dynamics, and The James Franck Institute, The University of Chicago, Chicago, United States

**Abstract** Specific protein-lipid interactions are critical for viral assembly. We present a molecular dynamics simulation study on the binding mechanism of the membrane targeting domain of HIV-1 Gag protein. The matrix (MA) domain drives Gag onto the plasma membrane through electrostatic interactions at its highly-basic-region (HBR), located near the myristoylated (Myr) N-terminus of the protein. Our study suggests Myr insertion is involved in the sorting of membrane lipids around the protein-binding site to prepare it for viral assembly. Our realistic membrane models confirm interactions with $PIP_2$ and PS lipids are highly favored around the HBR and are strong enough to keep the protein bound even without Myr insertion. We characterized Myr insertion events from microsecond trajectories and examined the membrane response upon initial membrane targeting by MA. Insertion events only occur with one of the membrane models, showing a combination of surface charge and internal membrane structure modulate this process.

## Introduction

Protein-lipid interactions play key roles in cell growth, signaling processes, and disease onset and propagation (*Khan et al., 2016*; *Whited and Johs, 2015*). Protein-protein interactions themselves are frequently dependent or mediated by specific lipids in the membrane (*Barros et al., 2016*; *Munro, 2002*); at the same time, they modify the local environment of the binding site to propagate a process or signaling cascade (*Monje-Galvan and Klauda, 2016*; *Sens et al., 2008*; *Stahelin, 2013*). Peripheral membrane proteins have received renewed attention over the past decade as they are involved in membrane deformation processes (*Simunovic et al., 2018*), lipid or small molecule transport between membranes (*Monje-Galvan and Klauda, 2016*; *Rogaski and Klauda, 2012*; *de la Ballina et al., 2020*), as well as viral assembly (*Sens et al., 2008*). Despite these advances, there are still unresolved questions in terms of the specificity of peripheral proteins to cell organelles, anionic lipids, and lipid rafts or domains (*Kerr et al., 2018*; *Olety and Ono, 2014*; *Sengupta et al., 2019*).

In the context of retroviral assembly, key peripheral proteins assemble at the membrane interface of an infected host, or can even start assembly in the cytosol and then target the plasma membrane (PM) for subsequent release (*Huber et al., 2017*; *Dick and Vogt, 2014*). Protein multimerization at the PM leads to the formation of an immature lattice and lateral reorganization on the membrane that results in the budding and release of an immature virion enclosing the viral genome (*Lalonde and Sundquist, 2012*). Given the relevance of protein interactions at the membrane interface for viral replication, it is important to understand the mechanisms of membrane targeting by proteins as well as the role of specific lipid species in protein binding and aggregation.

One of the most studied retroviruses is the human immunodeficiency virus type 1 (HIV-1). A key constituent of this virus is the group-specific antigen (Gag) polyprotein, which targets the PM and initiates the viral assembly process upon oligomerization at this interface (*Dick and Vogt, 2014*). Membrane targeting of Gag is mediated by its N-terminus, the matrix (MA) domain (*Mercredi et al., 2016*) – see *Figure 1*. The lipidated tail of MA, a myristoyl (Myr), is conserved

*For correspondence:
gavoth@uchicago.edu

**Competing interests:** The authors declare that no competing interests exist.

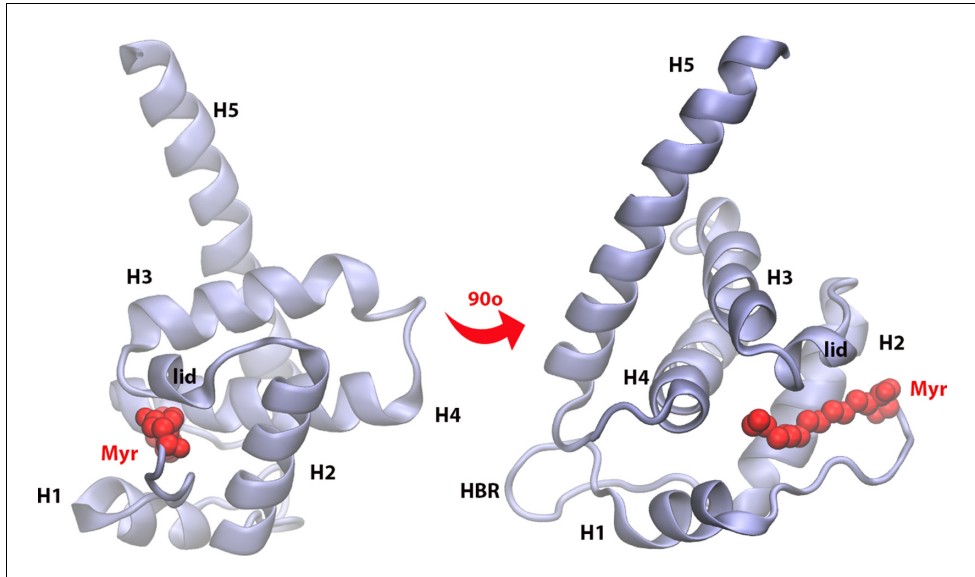

**Figure 1.** Matrix (MA) domain of HIV-1 Gag (residues 1–120) with key regions labeled: H1, H2, H3, H4, and H5 are helices numbered in order of appearance in the protein sequence; Myr is the lipidated tail of MA attached covalently to the N-terminus of the protein (shown in red); the lid is a small helical region that locks Myr inside the protein hydrophobic pocket; the HBR is a coiled region rich in LYS and ARG residues that is the main contributor to electrostatic interactions with the membrane.

across most retroviruses and is proposed to be key for interactions with lipids in the PM (*Vlach and Saad, 2015*). Several studies have examined and provided key conclusions in terms of protein-protein interactions to characterize the membrane targeting, viral assembly, and budding processes (*Lalonde and Sundquist, 2012*; *Tedbury et al., 2016*; *Yandrapalli et al., 2016*; *O'Carroll et al., 2013*; *Alfadhli et al., 2009*).

Among the main challenges in the study of membrane targeting domains, like MA, is the transient nature of their interaction and conformational changes at the membrane interface (*Eells et al., 2017*). Advances in experimental and computational techniques have enabled the study of such interactions in the context of viral replication (*Barros et al., 2016*; *Mercredi et al., 2016*; *Vlach and Saad, 2015*; *Pak et al., 2017*). In recent years, the specificity of the MA domain to the inner leaflet of the PM has been studied, showing that MA binds the PM through electrostatic and hydrophobic interactions mainly from its highly-basic region (HBR), located near the Myr site (*Vlach and Saad, 2015*). Different factors in membrane character and composition have been identified to enhance or reduce MA-binding affinity to the bilayer (*Barros et al., 2016*). For example, MA-PM interactions are far more complex than mere lipid headgroup recognition, and protein binding also modifies membrane dynamics (*Dick and Vogt, 2014*). MA aggregation is needed for the recruitment of lipids and other viral proteins to the viral assembly site in HIV-1 (*Sengupta et al., 2019*; *Leung et al., 2008*; *Metzner et al., 2008*). Additionally, the role of lipid rafts in protein-membrane interactions has been explored (*Lorent and Levental, 2015*). Nonetheless, the specific mechanism and conformational changes of MA that enable protein binding and aggregation are still unclear. The mechanism of Myr exposure and insertion as it pertains to the dynamics of the viral assembly site at the PM is not yet well understood. Finally, the effect of MA binding on lipid reorganization at the PM and how that renders the binding site more suitable for the recruitment of additional protein units of the viral bud has not been characterized in detail (*Dick and Vogt, 2014*; *Saliba et al., 2015*).

Here, we summarize an all-atom molecular dynamics (MD) study designed to gain molecular insight into the MA-membrane interactions. Given that lipid composition determines the character of a bilayer, and in turn, protein interaction and function (*Monje-Galvan and Klauda, 2015*), the study of processes at the membrane interface requires appropriate models to mimic the membrane environment, surface charge, and topology. We designed three symmetric membrane bilayers to model different aspects of the PM mimicking the composition of experimental studies of HIV-1 Gag

(*Yandrapalli et al., 2016*). Our models contain between three and six different lipid types at relevant ratios to model the PM in terms of charge, lipid packing, hydrophobic core structure, and to probe the effect of sterols in protein binding. The following sections report key conclusions about MA binding and Myr insertion mechanisms as well as detailed account of our simulation methodology.

## Results

All-atom MD simulations serve as a molecular lens to examine specific interactions between biomolecules. We built realistic membrane models to characterize the interaction between the MA protein of HIV-1 Gag polyprotein and lipids in the PM. We generated over 40 μs of trajectory, capturing for the first time the natural (unbiased) insertion of Myr, the lipidated tail of MA, into membrane bilayers. Additionally, our results highlight the relevance of selecting appropriate membrane models to study biological mechanisms. Insertion of Myr occurred repeatedly only into the membrane model built after the lipid composition of the inner leaflet of the PM, which we denoted as the *inner* model. We further characterized the effect of an inserted Myr tail in the other two membrane models, one built to reflect the ability of the PM to form lipid rafts (*raft*) and the other with no cholesterol to serve as a control model (*control*). However, natural insertion only occurred with our *inner* membrane model. Addition of multiple MA units did not result in Myr insertion events in the *control* or *raft* models, but did enhance the changes in the local membrane environment of the *inner* model.

In the following paragraphs, we discuss the MA-bound conformations observed in our trajectories and describe in detail the protein motions during binding and Myr insertion. We show HBR-membrane contacts are stable and strong enough to keep MA bound to all our membrane models, even without insertion of the Myr tail. Finally, we quantify the membrane response in terms of lipid tail order comparing systems with one or three MA units bound to the membrane vs. membrane-only systems.

To differentiate among the systems in this study, the following subscripts to the membrane models have been selected as notation: '*200*' for the 200 ns trajectories (e.g. $raft_{200}$, see also *Table 1*); '*pre*' for the trajectories starting from a pre-inserted Myr configuration (e.g. $raft_{pre}$, see also Table S1 in *Supplementary file 1*); '*myr*' for the system starting with a partially inserted Myr ($inner_{myr}$); '*mono*' and '*trimer*' for the systems simulating separated MA units vs. a formed trimer, respectively; the suffixes '1' and '2' to denote replicates; and the suffix 'L' to denote larger membranes with three MA units on the surface.

**Table 1.** Simulation components for the membrane models used in this study.
Chemical structures of each lipid species are shown in *Figure 2—figure supplement 1*.

| Model | Lipid species | Lipid mol fract. | Charge density ($e^-/nm^2$) | Unsat. degree | # Waters (small) | Ions | # Waters (large) | Ions |
|---|---|---|---|---|---|---|---|---|
| Control<br>$APL^*$ 66.9 +/- 0.84 | DOPC<br>DOPS<br>$PIP_2^{\ddagger}$ | 0.80<br>0.15<br>0.05 | −0.25 | 0.98 | 23442<br>*78.1*<br>$H\text{-}num^{\dagger}$ | 211 K<br>112 Cl | -<br> | -<br> |
| Inner<br>APL 47.8 +/- 0.40 | Chol<br>DOPC<br>DOPS<br>POPE<br>BSM<br>$PIP_2$ | 0.30<br>0.17<br>0.17<br>0.25<br>0.08<br>0.02 | −0.26 | 0.52 | 16009<br>*(53.4)*<br>H-num | 154 K<br>81 Cl | 103544<br>*(86.3)*<br>H-num | 295 K |
| Raft<br>APL 47.1 +/- 0.40 | Chol<br>DOPC<br>DOPS<br>BSM<br>$PIP_2$ | 0.28<br>0.30<br>0.06<br>0.30<br>0.06 | −0.32 | 0.54 | 15744<br>*(52.6)*<br>H-num | 170 K<br>83 Cl | 100034<br>*(83.4)*<br>H-num | 351 K |

*Area per lipid ($Å^2$/lipid).

†ydration number (# waters/lipid).

‡SAPI25 in CHARMM36m topology (18:0 - 20:4).

## MA-bound conformations

Two bound conformations, mouth-down (*blocked*) and mouth-to-the-side (*open*), occur within the first 100 ns of the short equilibration runs that started with MA at least 1 nm away from the membrane (see *Figure 2*). The protein was positioned with its mouth pointing downwards and Myr sequestered in the hydrophobic cavity of the protein; *Table 1* summarizes the details of the systems presented in this study. The protein was free to move in the solvent and the final bound conformation occurs indistinctly to the top or bottom leaflets of our symmetric models. The initial conformation of the protein did not influence the final bound state. *Video 1* shows an example where MA bound in the open conformation, the HBR loop interacts with the membrane and stablishes a permanent bound state from which Myr can exit the protein's hydrophobic cavity. Table S1 in *Supplementary file 1* lists the time of first contact with the bilayer as well as that of Myr exposure for the systems bound in the open conformation. Initial contact times with the *raft* model are approximately half than with the *control* or the *inner* models, and both bound conformations occur in the replica runs.

DOPC: 1,2-dioleoyl-sn-glycero-3-phosphocholine. DOPS: 1,2-dioleoyl-sn-glycero-3-phospho-L-serine. POPE: 1-palmitoyl-2-oleoyl-sn-glycero-3-phosphoethanolamine. BSM: N-octadecanoyl-D-erythro-sphingosylphosphorylcholine, a.k.a.(porcine) brain sphingomyelin. $PIP_2$: 1,2-dioctanoyl-sn-glycero-3-phospho-(1'-myo-inositol-4',5'-biphosphate).

In *Figure 2*, we illustrate the open and blocked bound conformations with the *control* and the *raft* models, respectively. In the open conformation, H1 and the HBR interact with the membrane surface and Myr is free to exit the protein's hydrophobic cavity, which occurs without much resistance. In fact, Myr is exposed and interacts with the lipid headgroups within our initial 200ns trajectories, though no insertion is observed from MA monomers until past the microsecond mark. Myr exposure occurs when the distance between the Lid and H1 regions, the mouth of the protein, opens up as shown in *Figure 2C* for the *control* and *raft* trajectories. On the other hand, the blocked conformation traps the Myr tail inside the protein hydrophobic cavity, which restrains its movement and prevents its insertion into the bilayer as it is discussed in subsequent sections. It is important to note the blocked conformation is not physically relevant when examining early stages of viral assembly, given the presence of the full Gag polyprotein would not allow for the interaction of H5 with the membrane as depicted in *Figure 2B*. The blocked conformation could, however, be a relevant structure of MA on the membrane surface during virus maturation, after budding and release of the virion and proteolytic cleavage of the MA-Capsid connection in the full Gag sequence (*Bukrinskaya, 2007*).

In both conformations, the electrostatic interactions are predominantly between LYS and ARG residues and DOPS or $PIP_2$ lipids in the bilayer, in agreement with experimental observations. The central panels in *Figure 2* show the lipid density of $PIP_2$ at the end of the short simulations. Interactions with anionic lipids occur mainly with the HBR in the open conformation (*Figure 2A control* model), and with H5 in the blocked conformation (*Figure 2B*, *raft* model). *Figure 2—figure supplement 2* shows the lipid density of DOPS at the end of the corresponding trajectories. Both $PIP_2$ and DOPS lipids co-localize to the protein-binding site in the open conformation, but initially only $PIP_2$ lipids interact with a protein bound in the blocked conformation. As discussed later, DOPS lipids interactions with the protein bound in the blocked conformation are slightly enhanced when Myr is inserted (see *Figure 2—figure supplements 2*, *3* and *4*). Finally, our micro-second trajectories show the protein is able to sample both bound conformations when Myr is (pre)inserted, partially modulated by contacts between the HBR and $PIP_2$ lipids (see *Figure 2—figure supplements 5* and *6*).

## Myristate insertion

Our simulations are the first to show unbiased reproducible Myr insertion events. *Table 2* lists the systems in which this process is observed, and Table S2 in *Supplementary file 1* lists the simulation length, protein-bound conformation, and initial Myr location for these systems. Myr insertionoccurred in two of the four trajectories with an MA monomer on the surface of a bilayer, and for multiple MA units in the simulation with a formed trimer. Systems with multiple MA units were run for shorter simulation times and insertion events occurred only in the system with an MA trimer within half of its simulated trajectory. None of the systems with three separate MA monomers exhibited Myr insertions.

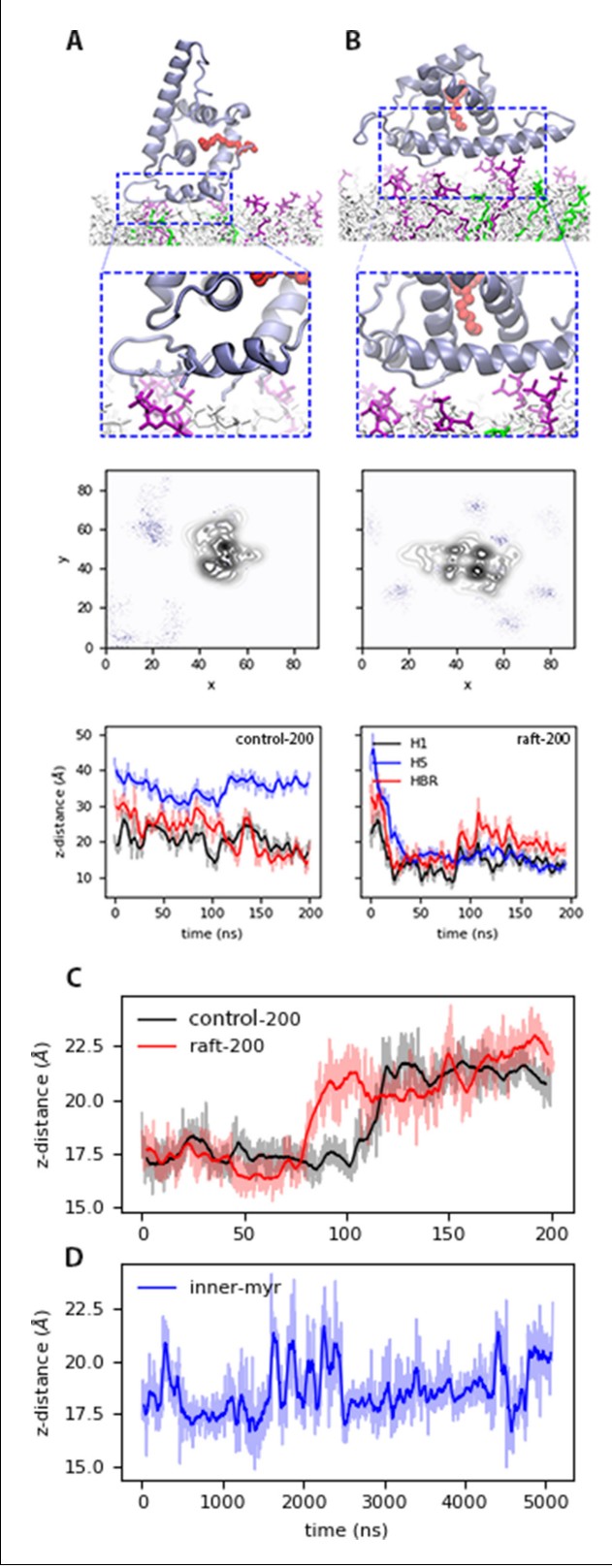

**Figure 2.** Bound states of MA with model membranes. (A) In the open state with the *control$_{200}$* model, and (B) the blocked state with the r*aft$_{200}$* model. *Top*: snapshots of the bound conformations, PIP$_2$ lipids are shown in purple and DOPS in green. *Center*: PIP$_2$ lipid density at the end of the 200-ns trajectories. *Bottom*: distance between protein sections and the phosphate region of the lipids in the binding leaflet for each conformation. Time series of

*Figure 2 continued on next page*

*Figure 2 continued*

the distance between H1 and Lid of MA, that is the mouth of the protein for the (C) *control*$_{200}$ and *raft*$_{200}$ systems with an open-bound and a blocked-bound MA, respectively. (D) H1-Lid distance for the *inner*$_{myr}$ model trajectory. Moving average for these plots is shown in bold lines every 5 ns for the 200-ns trajectories, and every 50 ns for the microsecond trajectory in bold lines; the faded curves are the original dataset. The chemical structures of each lipid species in the membrane models is shown in *Figure 2—figure supplement 1*.

The online version of this article includes the following source data, source code and figure supplement(s) for figure 2:

**Source code 1.** Python script to generate the bottom density plot of panels A and B in *Figure 2*.
**Source data 1.** Data to generate the bottom density plot of panels A in *Figure 2*.
**Source data 2.** Data to generate the bottom density plot of panels B in *Figure 2*.
**Figure supplement 1.** Chemical structures of the lipids used in the membrane models for this study.
**Figure supplement 2.** Lipid densities of DOPS lipids in the binding leaflets of the symmetric membrane models at the end of 200 ns of equilibration.
**Figure supplement 2—source code 1.** Python script to generate panels A, B, and C in the corresponding figure.
**Figure supplement 2—source data 1.** XY coordinates to plot the lipid density for DOPS in the binding leaflet.
**Figure supplement 3.** Protein-lipid contacts for the *control* models with pre-inserted Myr (*control*$_{pre}$, *control-1*$_{pre}$, *control-2*$_{pre}$).
**Figure supplement 4.** Protein-lipid contacts for the *raft, raft-1* and *raft*$_{pre}$ trajectories.
**Figure supplement 5.** Snapshots of the protein-membrane interactions for the *raft*$_{pre}$ model trajectory (pre-inserted Myr).
**Figure supplement 6.** Timeseries for the distance between H1, H5, and the HBR and the bilayer center for the *raft*$_{pre}$ system.

From our analysis, we propose an unrolling mechanism for Myr insertion into the binding leaflet. In all cases, insertion occurs only from the open bound conformation, that is when the H1 and HBR interact with the bilayer and H5 is extended towards the solvent. Myr insertion never takes places from a blocked bound conformation, in which the lipid tail can roll on itself and up and down the hydrophobic cavity of the protein, but cannot dive into the binding leaflet – that is insert C14 first into the leaflet. Importantly, insertion events were only observed into the membrane model with lipid composition characteristic to the inner leaflet of the PM, where viral assembly occurs in the biological system. *Video 2* shows a portion of the trajectory in which Myr is outside the hydrophobic cavity of the protein, interacts with the solvent and lipid headgroups, and then inserts into the bilayer. Myr insertion is never observed with the *raft* models, not even from the open bound conformations of MA monomers on the surface of large membrane patches, or from a formed MA trimer.

*Figure 3A* shows a series of snapshots showcasing the insertion of Myr into the *inner* membrane model, a process that occurs between 70 and 180 ns when a single MA protein is present. The first carbon in the tail, C2, interacts closely with the lipid headgroups upon exposure from its hydrophobic cavity; the tail can lay flat on the membrane surface while it searches for a large-enough lipid packing defect suitable for insertion. During insertion, the last carbon in the tail, C14, can extend vertically towards the water and pass by C2 as the tail rolls into the binding leaflet. Upon insertion, Myr remains extended

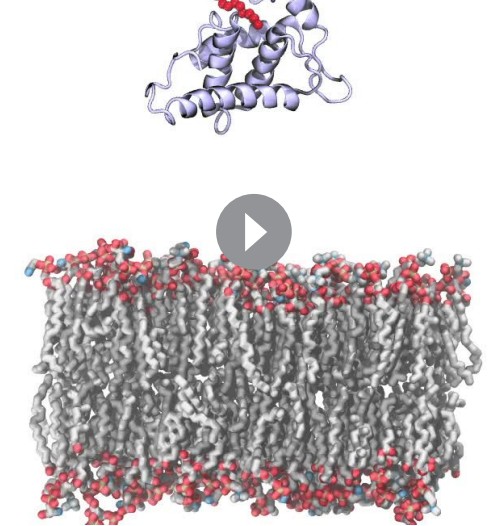

**Video 1.** MA binding in the open conformation. The protein binds within the first 50 ns of simulation trajectory; electrostatic interactions with charged lipid headgroups are strong enough to keep the protein bound to the membrane surface, and Myr is free to survey the membrane surface.
https://elifesciences.org/articles/58621#video1

**Table 2.** Summary of insertion events for the systems that exhibit Myr insertion.
The angle reported in the last column is that between the Myr tail and the bilayer normal, taken as the positive z-axis. The standard error is reported along with the angles.

| System | Sim time (μs) | Bound conf. | Bound leaflet | insertion (ns) | Myr angle (°) |
|---|---|---|---|---|---|
| Inner | 5 | Open | Bottom | $1380 \pm 2$ | $25.4 \pm 2.3$ |
| Inner-1 | 1.6 | Open | Bottom | $78 \pm 1$ | $8.2 \pm 1.4$ |
| Inner-L$_{trimer}$ | 1 | Open | Top | MA$_1$: $350 \pm 4$<br>MA$_2$: $780 \pm 2$<br>MA$_3$: *none* | MA$_1$: $37.8 \pm 5.4$<br>MA$_2$: $18.1 \pm 3.2$<br>MA$_3$: *N/A* |

and aligned to the other lipid tails in the bilayer core, and C14 is located at the center of the bilayer. The mechanism is similar when a formed MA trimer is bound to the bilayer, but in this case, Myr can also rest flat between the protein and the membrane, and roll into the leaflet from this position. Again, Myr insertion never occurs in a diving fashion (C14 first). Insertion is permanent on the time-scale of our simulations; however, the Myr tail can survey the local lipid environment, insert partially and exit again before permanently entering the bilayer. Such is the case of one of the MA monomers from the *inner-L$_{trimer}$* simulation (see *myr$_1$* in **Figure 3F**). This suggests a close relationship and feed-back between protein-lipid and lipid-lipid interactions. Myr 'sensing' could be a mechanism of lipid recruitment in preparation for viral assembly.

**Figure 3B** shows the root-mean-squared-fluctuations (RMSF) of the protein before, during and after Myr insertion for the *inner*-MA system. Note that residues 2–7, 19–29, 44–46, 52, 69–71, and 90–95 fluctuate the most, as is expected for coiled regions. However, residues 8–18 that correspond to H1, fluctuate more before and during Myr insertion, and stabilize afterwards. As Myr inserts, H1 gets closer to the bilayer and lays flat on the membrane surface; as observed in the time series inset on **Figure 3C** that shows the self-distance of both H1 and HBR computed as the z-component of the distance between the first and last alpha-carbons of each region (refer to Table S3 in **Supplementary file 1** for this metric). Upon insertion, Myr aligns to the lipid tails and remains in this position until the end of the simulation. **Figure 3D** shows the probability distribution of the cosine of the angle between Myr and the bilayer normal; in all cases, the lipid tail remains aligned to the z-axis inside the bilayer ($\cos(\theta)=1$), but C14 can retract to the mid-region of the leaflet occasionally.

To further examine the effects of Myr on the binding leaflet, we simulated an MA monomer on the surface of the small *inner* and *raft* membrane models starting from a pre-inserted Myr configuration. We ran three replicas of this configuration with the *control* model to assess the effect of cholesterol, that is the mechanical and structural properties of the membrane, on Myr retention and insertion. **Figure 3—figure supplement 1** shows the probability distribution of the cosine of the angle between Myr and the z-axis for these systems. Note that in a membrane environment rich in cholesterol and sphingomyelin lipids like the *raft* model, the angle distribution is much narrower than in the more fluid *control* model. This shows membrane structure and mechanical properties, modulated by its lipid content, can restrain the motion and dynamics of Myr. Contrary to our expectations, a partially inserted Myr in the *inner* model returned to the hydrophobic cavity of the protein; showing there is a much more complicated feedback loop between local lipid content and Myr insertion events. This observation leads us

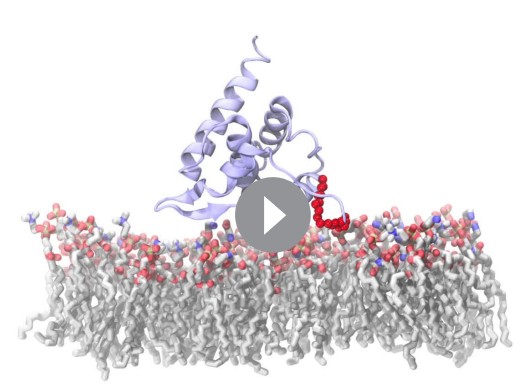

**Video 2.** 'Myr insertion into a model membrane. Exposure of Myr, the lipid tail of MA, occurs within the first 200 ns of trajectory. The tail can survey the membrane surface and even extend towards the water before permanently inserting into the bilayer.
https://elifesciences.org/articles/58621#video2

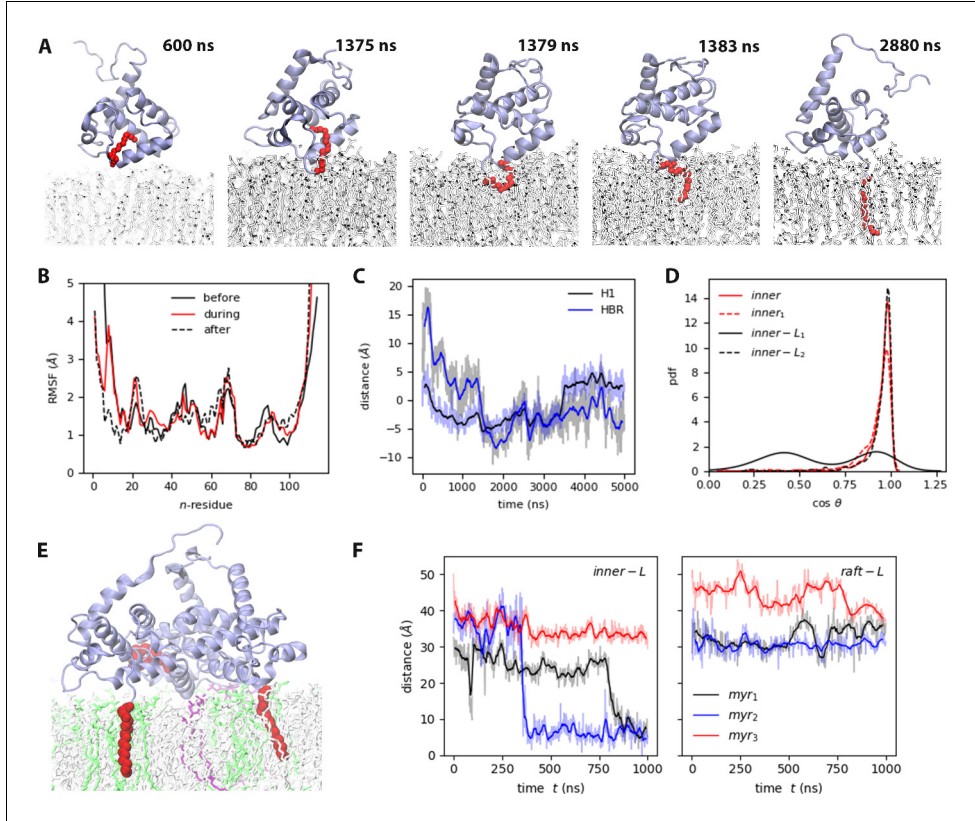

**Figure 3.** Myr insertion mechanism. (**A**) Trajectory snapshots from the *inner* trajectory, see also *Video 2*. (**A**) Myristate insertion mechanism (from the *inner* trajectory), see also *Video 2*. (**B**) RMSF of MA before, during, and after Myr insertion, at least 50 ns of the respective sections in the *inner* trajectory were used for this analysis. (**C**) Self-distance of H1 and HBR during Myr insertion (refer to Table S3 in *Supplementary file 1* for the definition of *self-distance*). (**D**) Probability distribution of the angle between Myr and the bilayer normal for the MA units that exhibit Myr insertion (see also *Figure 3—figure supplement 1*). (**E**) Snapshot of the side view of *inner-L$_{trimer}$* at the end of the simulation, PIP$_2$ lipids are shown in purple, DOPS lipids in green. (**F**) Time series of Myr insertion in the first and second MA units in the *inner-L$_{trimer}$* simulation, the distance was computed between the last carbon in the lipid tail (**C14**) and the bilayer center. All the time series show the blocked moving averaged in solid color and the raw data faded.

The online version of this article includes the following figure supplement(s) for figure 3:

**Figure supplement 1.** Probability distribution of the cosine of the angle between Myr and the bilayer normal (z-axis) in the systems starting from a pre-inserted Myr configuration.

to suggest Myr may act as a lipid sensing probe during viral assembly and contribute to the recruitment of lipids and other proteins to the viral assembly site.

A closer look at the carbon atoms in the Myr tail during insertion shows C2, the first carbon in the Myr tail attached to the GLY residue in the N-terminus of MA, does not shift much after Myr exposure and insertion (see top panels in *Figure 4*). As such, the probability density of C2 has only one peak in the cases with a single MA unit on the membrane (panels A and B in *Figure 4*). *Figure 4*.C suggests C2 can go deeper into the bilayer after insertion of Myr into the bilayer core when multiple units are present, resulting in two peaks in its probability density. This may occur as a cumulative effect of having multiple Myr insertions from MA clusters; as more Gag units bind to the PM, one could imagine more Myr insertions push the forming protein platform a little deeper into the bilayer vs. single protein units in the membrane plane.

The last carbon of Myr, C14, is located very close to the bilayer center upon insertion; since this is a shorter carbon tail compared to the 16 to 18 carbons of the majority of the bilayer lipids, it is understandable C14 remains on average around 0.6 nm away from the bilayer center. Interestingly, when Myr returns to the hydrophobic cavity, it lays nearly horizontally as in *Figure 4B*, aligned to the

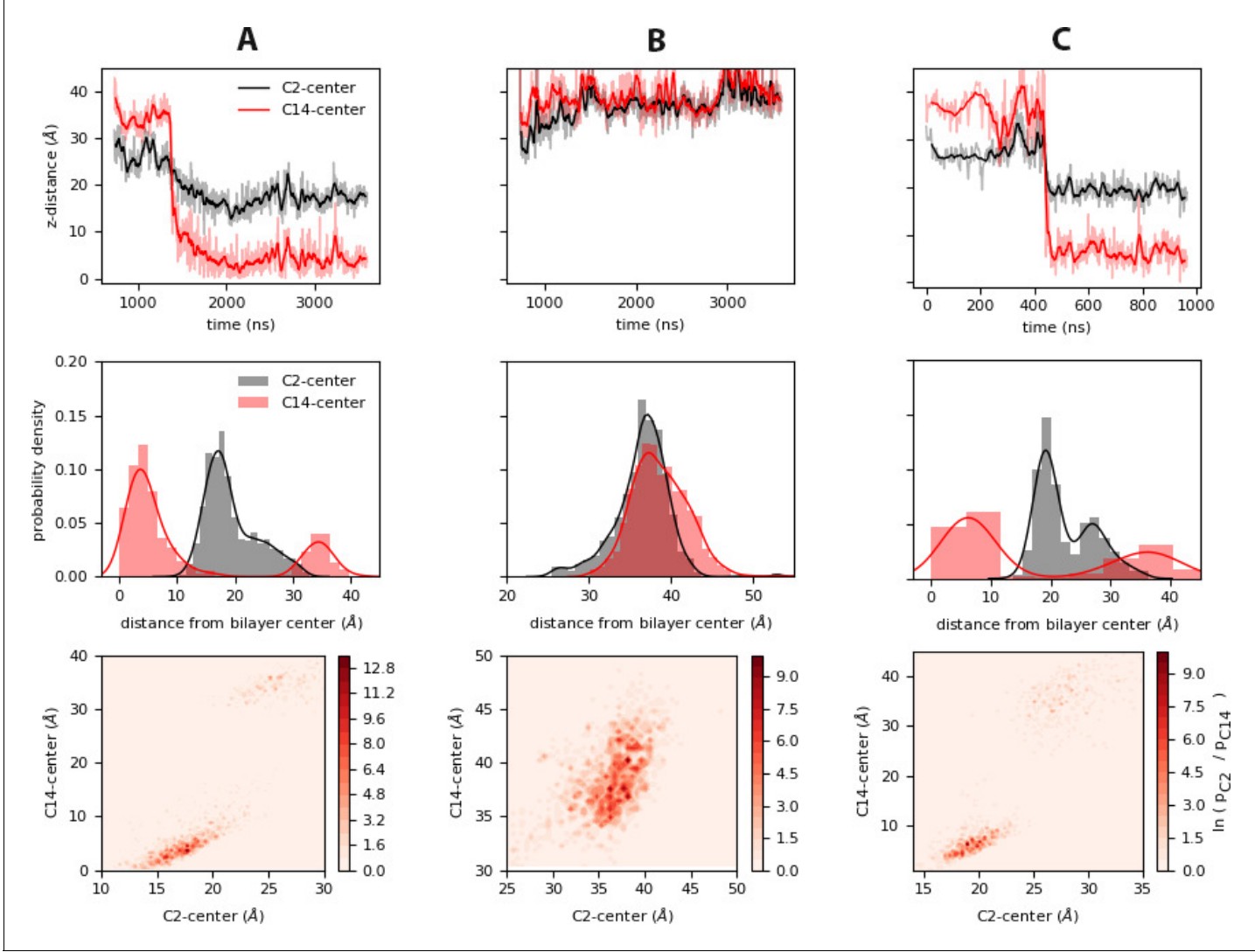

**Figure 4.** Relative motions of the Myr tail during insertion. *Top*: Time series of insertion for the first and last carbons in the Myr tail, C2 and C14; the distance was computed between the center of mass of the respective carbon and the bilayer center as estimated from the average distance between the phosphate regions of the lipids in each leaflet. The faded regions show the raw data of this metric, and the bolded regions show the corresponding blocked moving average. *Center*: Probability density of the same distances as estimated from the normalized histograms of each time series. *Bottom*: Heat maps of the probabilities of the C2 and C14 atoms of Myr as estimated from the histogrammed data. The systems shown are: (**A**) *inner*, (**B**) *inner_myr*, and (**C**) *inner-L_trimer* (MA_2 unit).

The online version of this article includes the following source data and source code for figure 4:

**Source code 1.** Python script to generate panel A; (top).
**Source code 2.** Python script to generate panel B; (top).
**Source code 3.** Python script to generate panel C; (top).
**Source data 1.** Data to generate panel A.
**Source data 2.** Data to generate panel B.
**Source data 3.** Data to generate panel C.

membrane surface inside the protein with C2 and C14 at the same height above the bilayer center. This is also observed in the probability distribution of the cosine of the angle between Myr and the z-axis for this system, included in *Figure 3—figure supplement 1* for comparison with the data for the *raft* and *control* simulations that started with a pre-inserted tail.

## Membrane response

MA binding results in co-localization of charged lipids to the protein binding site, even when only a single unit is present (refer to *Figure 2* and *Figure 2—figure supplement 2*). PIP$_2$ recruitment is not required for Myr insertion, but in some instances the number of contacts with MA increase upon Myr insertion, like with the *inner* trajectories shown in *Figure 5*, and the *raft* system in *Figure 2—figure supplement 3*. In other cases, the number of contacts between MA and PIP$_2$ remains the same or increases even without insertion of the lipid tail, as in the trajectories with the *raft* model where the protein is bound in the blocked conformation (see *Figure 2—figure supplement 4*). Note that MA can still move on the membrane surface after its initial binding and prior to Myr insertion, but is constricted to a given local lipid environment afterwards, as shown in the simulations starting with a pre-inserted Myr. For these latter cases, a maximum of 57% or 44% of PIP$_2$ present colocalizes to the protein binding site in the *control* and *raft* models, respectively, whereas 75–100% of PIP$_2$ lipids bind to the protein in the *inner* model trajectories.

In addition to PIP$_2$, DOPS lipids also interact with MA; these are present in higher concentration in the *control* and *inner* models, but at the same concentration as PIP$_2$ in the *raft* model. When Myr is pre-inserted, the interaction of PS lipids with the protein increases from 12 to up to 30% of its content in the *raft* model, but fluctuates greatly between 9–35% in the *control* model, potentially due to the more fluid nature of this model without cholesterol. From the trajectories with the *inner* model, PS-protein interactions are not affected by Myr insertion, and a maximum of 23% of PS present in the membrane interacts with the protein throughout the simulation. Notably, a maximum of 38% of PS interacts with the protein for the simulation with the *inner* model that started with a partially inserted Myr, which decreases to 19% after Myr returns to the protein's hydrophobic cavity. Again, these specific interactions show there is a tight loop between Myr insertion events and local lipid composition.

Increasing protein content on the *inner-L$_{trimer}$* model results in enhaced MA-DOPS interactions after Myr insertion (see *Figure 5*); yet, PIP$_2$ interactions do not follow a steady pattern. This does not occur on the *Raft-L$_{trimer}$* model, where electrostatic interactions between the MA trimer and the membrane keep the proteins bound, but there are no Myr insertion events. In this case, the interactions with anionic lipids are in fact reduced to nearly 50%. *Figure 5—figure supplement 1* shows lipid density plots for PIP$_2$ on the membrane plane along with the relative locations of MA monomers or an MA trimer on the surface. In the case of the free monomers, PIP$_2$ co-localizes to the protein binding site in the same fashion as observed in the simulations of a single MA unit. As the monomers interact with each other, PIP$_2$ density further increases locally. Although the monomers do not form a trimer in the length of our simulations, it is clear that MA-MA interactions modifies the local lipid concentration, and recruits specifically PIP$_2$ lipids to the protein binding site, enhancing it for viral assembly.

As a final metric to characterize membrane response, we computed the deuterium order parameters (S$_{CD}$), a measure of order in terms of the alignment of lipid tails inside the bilayer. We report the difference in lipid order between the systems exhibiting Myr insertion on a given membrane model vs. the membrane-only simulation of the same model. The S$_{CD}$ values were computed per leaflet after Myr insertion for the protein-membrane systems, distinguishing between the binding (BL) and the opposite leaflets (OL). The difference between the S$_{CD}$ value of the protein-membrane and membrane-only systems is shown in *Figure 6*. A positive number indicates the protein-membrane system has higher S$_{CD}$ values, that is the lipids tails are more ordered inside the bilayer, while a negative value indicates more disorder in the lipid tails when the proteins are present and Myr is inserted.

The long tails of PIP$_2$ and sphingomyelin lipids, with 20 and 22 carbons respectively, experience an increase in order when Myr inserts into the bilayers. However, the trend is different when an MA trimer is present instead of a monomer. This is particularly noticeable for the sphingomyelin values, which show an increase in order in the middle section of the lipid tail in the binding leaflet when a trimer is present instead of a monomer. Given the ability of Myr to survey the lipid environment prior to stable insertion, this increase in order in the middle of the binding leaflet could potentially be a way to recruit additional MA units to the binding site. On the other hand, the sphingomyelin lipids in the opposite leaflet show an even larger effect of disorder in the tail region, near the bilayer center. Although out of the scope of this study, these results indicate there may be a modulated response

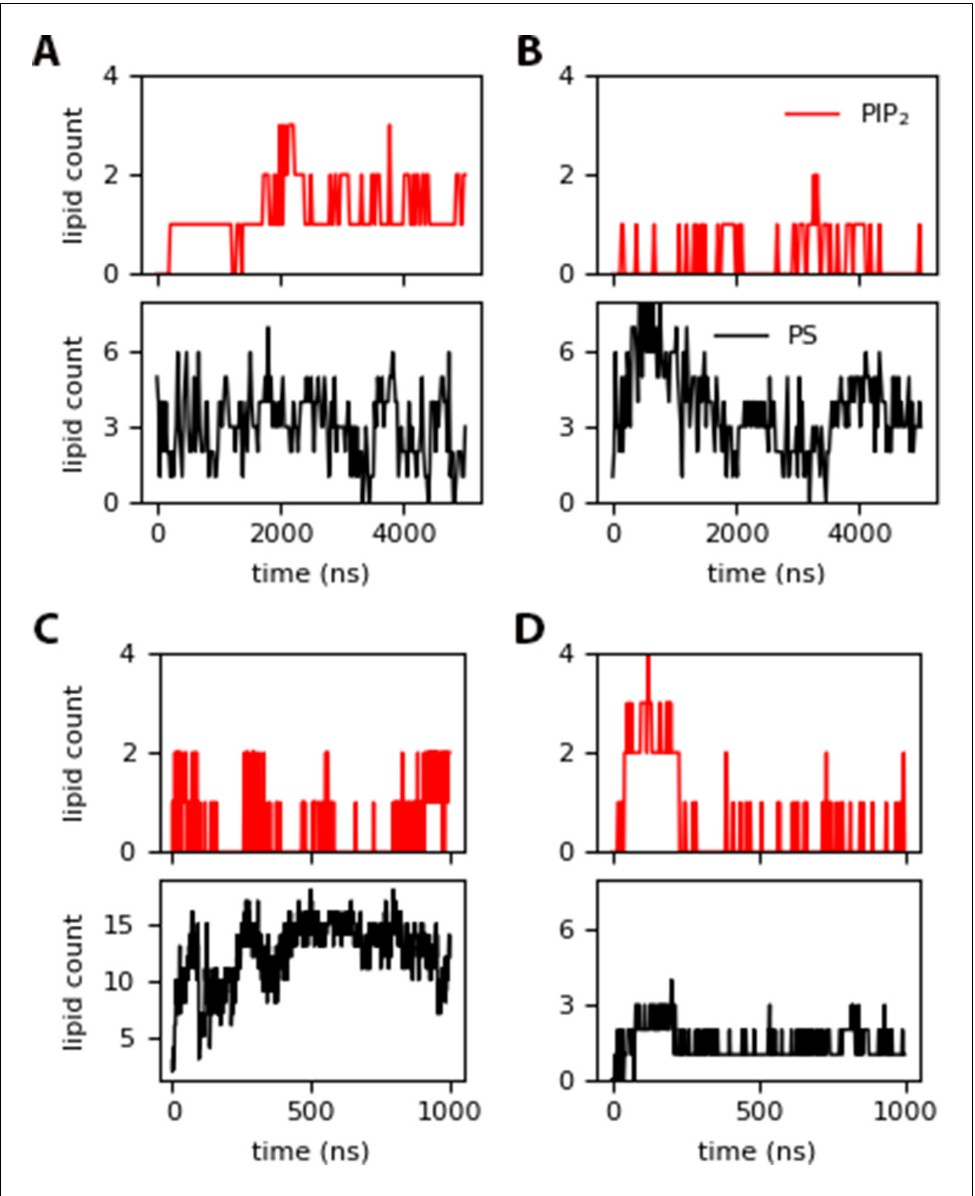

**Figure 5.** PIP$_2$ and DOPS contacts within 10  of the protein for: (A) *inner* (insertion around 1380 ns), (B) *inner$_{myr}$*, (C) *inner-L$_{trimer}$* (with permanent Myr insertion after 400 ns), and (D) *raft-L$_{trimer}$* (with no insertion).

The online version of this article includes the following source data, source code and figure supplement(s) for figure 5:

**Figure supplement 1.** PIP$_2$ lipid densities in the binding leaflet for the: (A) *inner-L$_{trimer}$* (B) *raft-L$_{trimer}$*, (C) *inner-L$_{mono}$* (D) *raft-L$_{mono}$* systems.

**Figure supplement 1—source code 1.** Python script to plot the lipid density for PIP$_2$ in the binding leaflet for the (A) *inner-L$_{trimer}$* (B) *raft-L$_{trimer}$*, (C) *inner-L$_{mono}$* (D) *raft-L$_{mono}$* systems.

**Figure supplement 1—source data 1.** XY coordinates to plot the lipid density for PIP$_2$ in the binding leaflet for the (A) *inner-L$_{trimer}$* (B) *raft-L$_{trimer}$*, (C) *inner-L$_{mono}$* (D) *raft-L$_{mono}$* systems.

**Figure supplement 2.** Lipid densities for sphingomyelin (top) and cholesterol (bottom) lipids in the binding and opposite leaflets, for the *inner-L$_{trimer}$* trajectory.

**Figure supplement 2—source code 1.** Python script to plot the lipid density for sphingomyelin and cholesterol in the binding and opposite leaflet, respectively.

**Figure supplement 2—source data 1.** XY coordinates to plot the lipid density for sphingomyelin and cholesterol in the binding and opposite leaflet.

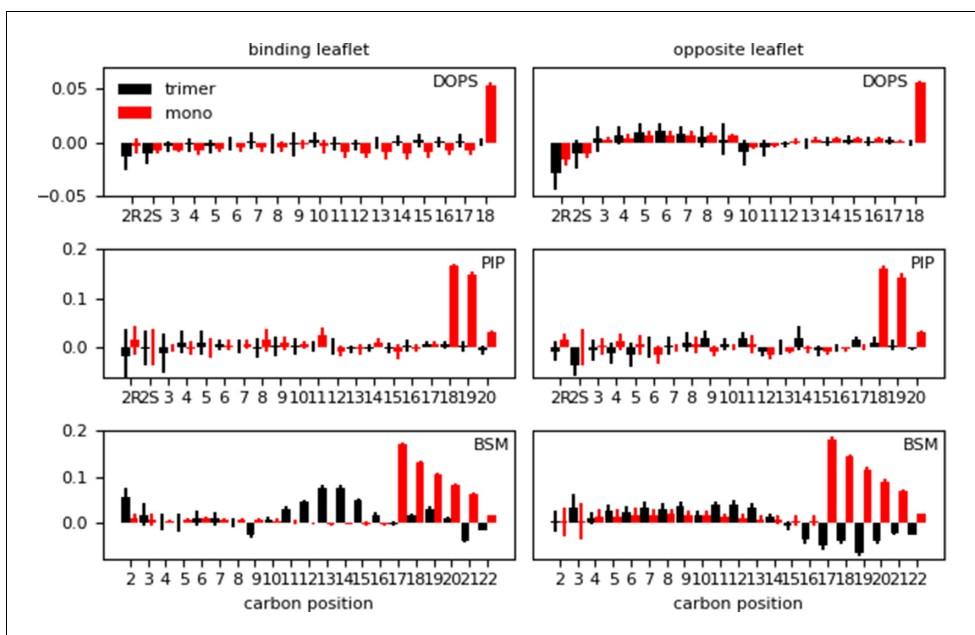

**Figure 6.** Difference in lipid order parameter ($S_{CD}$) between the protein-membrane and membrane-only systems per leaflet for the *inner* ('mono') and *inner-L$_{trimer}$* ('trimer') systems after Myr insertion. A positive value indicates the presence of the protein with inserted Myr increases the order in the bilayer, whereas a negative value corresponds to a decrease in order with respect to the membrane-only system.

to protein binding and Myr insertion between the PM leaflets, which has also been suggested by other groups (*Sengupta et al., 2019*; *Dick and Vogt, 2014*; *Dumas and Haanappel, 2017*).

## Further analysis and discussion

We examined the effect of membrane composition on the binding mechanism and insertion of the lipidated tail of MA, the membrane targeting domain of HIV-1 Gag polyprotein. Using MD, we simulated MA monomers as well as formed trimer near the surface of different membrane models. These models were selected based on experimental studies that also examined the interplay between protein binding and aggregation on membranes in the context of viral assembly (*Yandrapalli et al., 2016*). Our models mimic the sterol content, surface charge, and key lipid species in the PM. Although we used a symmetric bilayer in this study, our models provide important insight on the protein-membrane interactions needed during the early stages of viral assembly. The key contributions from our work include specific protein-lipid interactions at the binding site, a molecular-level mechanism for Myr insertion, characterization of protein conformational changes that enable Myr insertion, and the local changes in the membrane as a result of binding and insertion.

In agreement with experimental studies, the composition of the membrane affects and modulates the interaction of MA with the bilayer (*Mercredi et al., 2016*; *Barros et al., 2016*; *Alfadhli and Barklis, 2014*; *Saad et al., 2006*). All our models have a minimum of 12% anionic character, which is enough to promote electrostatic interactions with the HBR or H5 regions of the protein depending on bound conformation. Most of these interactions occur between LYS or ARG residues and PIP$_2$ and DOPS lipids, as has also been stated in the literature (*Mercredi et al., 2016*; *Eells et al., 2017*; *Saad et al., 2006*). *Figure 2* and *Figure 2—figure supplements 2–4* show enrichment of PIP$_2$ at the protein-binding site as it has been suggested several times before (*O'Carroll et al., 2013*; *Alfadhli et al., 2009*; *Barros et al., 2016*; *Charlier et al., 2014*). Upon initial contact and binding, the protein is relatively free to search the membrane surface, as we observed on our microsecond trajectories with the *inner* and *raft* models with both MA monomers and a formed trimer. Insertion of the lipidated tail restricts the movement of MA, although it does not serve as the only anchor because the protein remains bound for the entirety of 3 or 5 µs trajectories even in the cases where Myr insertion is not observed (*raft* model). Myr has been suggested as an anchor before, but

experimental studies with Myr-less MA mutants also observed MA binding to membranes (*Eells et al., 2017*; *Barros et al., 2016*). So far, there has not been a unified answer as to what is the role of Myr in MA membrane targeting and subsequent oligomerization of Gag at the inner leaflet of the PM. Below we propose specific MA-membrane interactions that enable exposure and insertion of Myr dependent on the membrane nature, which has not been captured through simulations previously.

We observe quick MA binding to our symmetric models in two conformations during the short simulation trajectories, shown in *Figure 2* and summarized on Table S1 in *Supplementary file 1*. Both conformations are stabilized mainly by electrostatic interactions with PIP$_2$ lipids, in agreement with various reports (*Mercredi et al., 2016*; *Charlier et al., 2014*). The protein orientation in the open conformation agrees with experimental and other simulation studies (*Eells et al., 2017*; *Charlier et al., 2014*). In this conformation H1 and HBR interact closely with the bilayer and H5 is extended towards the solvent; this is the only conformation from the two observed in our simulations that is plausible during early stages of viral assembly when the full Gag protein is present and connected to MA after H5. This conformation allows for intermittent interaction of H2 with the bilayer and seems to be the preferred orientation of MA. The open conformation is strongly modulated by HBR-PIP$_2$ interactions, as shown in *Figure 2—figure supplement 5* for one trajectory with the *raft$_{pre}$* model. In this example, Myr started from a pre-inserted configuration and the protein bound in the blocked conformation, that is H5 flat on the membrane surface and HBR pointing toward the solvent. PIP$_2$-mediated interactions with the HBR and a shift of H1, attached to Myr, into the lipid head-group region enable the permanent transition from the blocked to the open conformation within the first microsecond of simulation until the end of the 5 µs trajectory (see *Figure 2—figure supplement 6* for the protein-membrane distance series of this system). It appears that the open bound conformation is the preferred orientation of MA, even without the presence of the full Gag in our studies.

Myr exposure is readily observed from the open conformation within the 200-ns equilibration trajectories. The lipid tail exits the protein's hydrophobic cavity as the Lid region separates from H1, opening the mouth of the protein. *Figure 2C and D* shows time series of this distance corresponding to the two systems shown in *Figure 1*. Opening of the Lid-H1 region occurs in both the open and blocked bound conformations of MA, but only the open conformation allows for Myr exposure. In the blocked conformation, opening of the mouth only results in more room for Myr to wiggle inside the hydrophobic cavity of the protein; this configuration would allow for Myr diving into the membrane, but this mechanism is not observed in any of our trajectories – the preferred insertion mechanism is discussed in the following paragraphs. Opening of the Lid-H1 region occurs both for exposure and sequestration of Myr; in the trajectory of MA with the *inner* model starting from a partially inserted Myr, we see the lipid tail return to the hydrophobic cavity (*Figure 4B*) as the mouth opens and closes to lock the tail back inside the protein (*Figure 2D*).

Previous studies examined the protein in solution and suggested MA trimerization was needed for Myr exposure (*Tang et al., 2004*). *Charlier et al., 2014* used umbrella sampling to identify a barrier close to 8 Kcal/mol for full Myr exposure in solvent. We conclude that the character of the membrane has a greater influence on protein-lipid interactions that allow Myr exposure and later insertion. In the short equilibrations, we performed for a single MA or the MA trimer in solvent prior to positioning them above equilibrated membrane coordinates, we actually observe Myr sequestration into the hydrophobic cavity of the protein, which is expected intuitively as the hydrophobic tail would prefer a hydrophobic environment over surrounding water. Interestingly, one of our short protein-only equilibration systems did result on Myr leaning in the outer hydrophobic cavity of the protein, formed on the outer surface of the protein between the Lid and H3. We did not use this structure in any of our protein-membrane simulations, but it would be an interesting test case to examine the protein-membrane interactions that enable 'release' of Myr from this external hydrophobic pocket. This external cavity is different from the location suggested from NMR studies to be a lodging pocket for membrane lipid tails to anchor MA to the membrane (*Vlach and Saad, 2015*; *Saad et al., 2006*), but suggests a lipid tail could potentially find an energetically favorable location outside the protein.

In itself, Myr exposure is not a determining step for insertion into the bilayer or protein retention on the membrane surface. We observe Myr insertion into our *inner* membrane model repeatedly in an unrolling fashion from an MA monomer or a formed trimer. In the trimer configuration, each protein unit is bound to the membrane in the open conformation, the mouth of the protein open side-

wise and nearly perpendicular to the bilayer. We simulated trimers on the surface of the *inner* and *raft* models; nonetheless, no insertion is observed onto the *raft* model. Examining the nature of the membrane models, we see lipid packing in the bilayer as measured by the surface are per lipid (APL) is not a key determinant for Myr insertion. As listed on *Table 1*, the average APL is not statistically different between the *inner* and *raft* models. Key differences between these membrane models are the surface charge, 19% on the *inner* model vs 12% on the *raft* model, sphingomyelin content, 8% vs 30% respectively, and POPE content, 25% vs none respectively. We did not quantify lipid packing defects on our systems, but by simple inspection of the lipid species in each model, the *raft* model is expected to have more tightly packed lipid headgroups on its surface given the relative size of the headgroups in the lipid species in this model vs those in the *inner* model (see *Figure 2—figure supplement 1*). Lastly, in the actual biological system, MA is not expected to bind a leaflet rich in cholesterol/sphingomyelin lipids, as is the outer leaflet of the PM. The fact Myr insertion is not observed onto the *raft* model also demonstrates the need for careful selection of lipids in the membrane models used to study protein interactions at the membrane surface, or inside the bilayer for that matter (*Monje-Galvan and Klauda, 2015*).

Considering the effect of Myr on lipid-lipid interactions, we observe sphingomyelin aggregates are more defined when Myr is inserted, for example, in the binding leaflet of the *raft$_{pre}$* system. Furthermore, cholesterol molecules co-localize to the edge of sphingomyelin regions in the binding leaflet, but not inside them. Interestingly, these enriched regions are not below the protein binding site. The effect on lipid enrichment is not as marked in the *inner* model runs with a single MA monomer given the sphingomyelin content is much lower than that of the *raft* model. Nonetheless, as shown in *Figure 5—figure supplement 2*, there are some sphingomyelin-cholesterol enriched regions in the *inner-L$_{trimer}$* trajectory. The presence of the trimer induces the formation of smaller or compact aggregates that are again outside of the protein-binding site. In this case, however, cholesterol molecules can aggregate below the protein itself. An interesting note is that Myr seems to avoid fully saturated regions for its initial insertion; *Figure 3F* shows the MA$_1$ unit in the trimer partially inserted into the bilayer and exit immediately to reinsert permanently later in the trajectory. The initial insertion site was rich in cholesterol molecules, which made the lipid tail bounce back into the lipid headgroup region. Permanent insertion occurred when less cholesterol molecules were present around the protein, yet sterol molecules later accumulated around Myr afterwards. The dynamics of Myr sensing for permanent insertion remain to be fully elucidated; our results, however, suggest a feedback loop between Myr insertion events and local lipid composition.

As expected, no lipid aggregates are observed in the *control* model because it lacks the lipids involved in domain/raft formation. Furthermore, these trajectories show no statistical difference in the lipid order parameters (S$_{CD}$) compared to the corresponding membrane-only system even starting from a pre-inserted Myr configuration. Whereas these values in the *inner* membrane show an increase in order when the protein is present and Myr is inserted in both the monomer and trimer trajectories (see *Figure 6*). Clearly, the membrane composition determines the extend of influence of the protein on lipid-lipid interactions. Capturing the changes in lipid dynamics on the membrane indicates Myr plays a role in lipid sorting and dynamics at the viral assembly site, rather that influencing the MA-binding events. We did not quantify lipid clustering or interleaflet dynamics in this study but have reserved this analysis for a future publication examining MA binding and aggregation on asymmetric bilayers.

As a final assessment, we characterized Myr insertion events using a reduction technique to quantify the slowest motions of the process. We performed time-independent component analysis (tICA) on the distances between each carbon of the Myr tail and the alpha-carbons of the structured regions of the protein as labeled on *Figure 1*: H1-5 and the HBR, no other coiled region was considered. *Figure 7* shows two cases with the *inner* membrane model; panel A is for the trajectory exhibiting Myr insertion (*inner* system) and panel B for the trajectory starting with a partially inserted Myr that resulted in its sequestration back into the protein's hydrophobic cavity (*inner$_{myr}$* system). Notice the first independent component (tIC) identified in the analysis seems to be rather related to protein motions for Myr exposure or sequestration, such as the opening and closing of the mouth of the protein, that is the H1-Lid distance (refer to *Figure 2D*). The second tIC captures Myr insertion, or lack thereof, depending on the trend of the time series. *Figure 7A* shows a concave shape for the second tIC that decreases around the time of Myr insertion; whereas panel B shows a convex shape

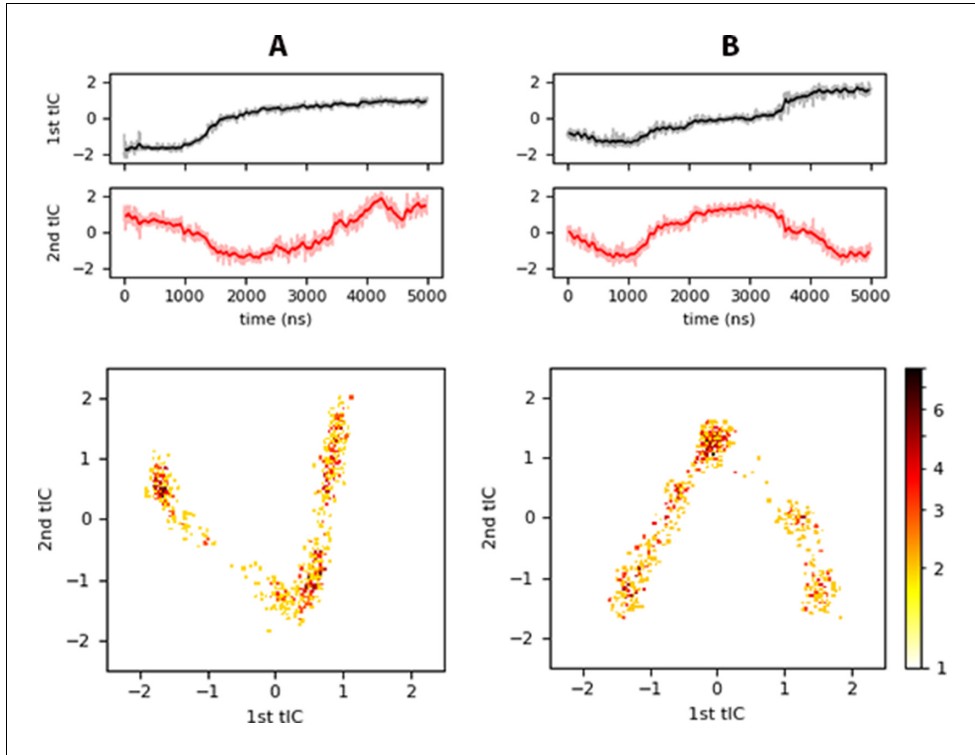

**Figure 7.** tICA results for (A) *inner*, and (B) *inner*$_{myr}$. *Top* panels show the projection of each trajectory onto the slowest independent components identified, and the *bottom* panels a heat map of histograms of the projected trajectory onto the respective tIC (logarithmic scale). Similar plots are shown in ***Figure 7—figure supplement 1*** for the *control*$_{pre}$, *raft*, and *raft*$_{pre}$ systems.

The online version of this article includes the following source data, source code and figure supplement(s) for figure 7:

**Source code 1.** Python script to generate panels A and B of ***Figure 7*** as well as A, B, and C of ***Figure 7—figure supplement 1***; timeseries of the trajectory projected onto the first and second tIC components.

**Source data 1.** Data to generate panel A of ***Figure 7***; timeseries of the trajectory projected onto the first and second tIC components.

**Source data 2.** Data to generate panel B of ***Figure 7***; timeseries of the trajectory projected onto the first and second tIC components.

**Figure supplement 1.** tICA results for (A) *control*$_{pre}$, (B) *raft*, and (C) *raft*$_{pre}$ trajectories.

**Figure supplement 1—source data 1.** Data for panel A of ***Figure 7—figure supplement 1***; timeseries of the trajectory projected onto the first and second tIC components.

**Figure supplement 1—source data 2.** Data for panel B of ***Figure 7—figure supplement 1***; timeseries of the trajectory projected onto the first and second tIC components.

**Figure supplement 1—source data 3.** Data for panel C of ***Figure 7—figure supplement 1***; timeseries of the trajectory projected onto the first and second tIC components.

for the second tIC that increases as Myr is sequestered back into the protein around 2 μs, same time in which the H1-Lid region opens to make way for the lipid tail (see ***Figure 2***.C and D).

The bottom panels in ***Figure 7*** are 2D histograms of the trajectory projections onto the slowest tICs in logarithmic scale for clarity. These show a V-shape characterizes Myr insertion, while an inverted V characterizes the motion of Myr away from the bilayer center, either to the protein cavity or remaining around the lipid-headgroup region. The same characteristic shapes are observed for the corresponding plots of the *control*$_{pre}$, *Raft*, and *Raft*$_{pre}$ systems in ***Figure 7—figure supplement 1***. In these cases, the systems with pre-inserted Myr (***Figure 7—figure supplement 1A & C***) show a first tIC that increases in time, and a second tIC with a concave shape. Again, systems with an inserted Myr have a 2D histogram with V-shape behavior, while the system with no insertion shows an inverted V. Simply looking at the first tIC across the *inner*, *raft*, and *control* models, it is apparent

the intra-protein and protein-membrane interactions are different depending on the membrane model in use. This observation further proves the need for realistic models to study protein and lipid interactions at or on membranes.

The coefficients to each eigenvector, known as the load of each tIC, show the relative importance of each $Cn_{Myr}$ - $C\alpha_{prot}$ distance for the process under study, Myr insertion in this case. These distances were deemed relevant because the respective normalized loads (coefficients) were higher than ±0.5. Looking only at the two slowest tICs, depicted in *Figure 7*, our analysis revealed the key distances during Myr insertion are between the first carbons in the tail (C2 and C3) and the HBR, followed by the distance between the same carbons and H5 or H1 for the 1st and 2nd tIC, respectively. Additional contributions to these tICs came from the distance between carbons C3, C6, and C13 with H5, and between carbon C14 and the HBR as well as H1. These observations provide further insight on key protein regions that contribute to the tail's insertion (the HBR, H1 and H5), and corroborate the preferred mechanism of Myr insertion by an unrolling of its tail.

## Materials and methods

Three membrane models were designed to study the binding dynamics of MA based on an experimental study from *Yandrapalli et al., 2016*. *Table 1* contains the lipid components of each bilayer along with other system details. The *control* model was selected to examine the impact of cholesterol on protein-membrane interactions. The *inner* model mimics the anionic character of the PM, while the *raft* model emphasizes the ability of the PM to phase separate and form lipid clusters enriched in sphingomyelin and cholesterol lipids. The *raft* model is not a model for the outer leaflet of the PM, but a membrane model containing raft-forming lipids and a charge density higher than the other models in these studies to ensure electrostatic binding of MA. Note that both the *inner* and *raft* models were built to reflect lipid content of the PM; the *inner* model has 25 mol% of PE lipids, known to reside mainly in the inner leaflet of the PM, and nearly three times the amount of DOPS lipids than the *raft* model. On the other hand, the *raft* model, built to reflect the existence of lipid rafts (or regions of high order) in the PM, lacks PE lipids but is rich in sphingomyelin and cholesterol (nearly 60 mol%), both the main components of rafts. $PIP_2$ and DOPS lipids are included to account for the anionic content of the membrane.

Fully hydrated bilayers (35+ water molecules per lipid) and an MA protein were built separately using CHARMM-GUI *Membrane Builder* and *Quick-Solvator*, respectively (*Jo et al., 2007*; *Jo et al., 2008*; *Brooks et al., 2009*; *Wu et al., 2014*; *Lee et al., 2016*). All systems were rendered electrostatically neutral using KCl salt at a 0.15 M concentration as detailed in *Table 1*. MD simulations were performed using the CHARMM36m force field (*Huang et al., 2017*), with its most up-to-date parameters for lipids and proteins. Initial equilibration of the bilayers and protein were performed using the GROMACS MD simulation package (*Abraham et al., 2015*), and micro-second production runs were carried on the Anton two machine (*Shaw et al., 2014*). Coordinates for an MA monomer were extracted from PDBID:1HIW in the protein data bank (https://www.rcsb.org) and a myristoyl (Myr) tail added to its N-terminus on CHARMM-GUI. Initial relaxation of the bilayer systems was carried out over 250 ps following the six-step protocol provided on CHARMM-GUI (*Jo et al., 2009*), and each system equilibrated for at least 100 ns on GROMACS before introducing the protein in the solvent region.

The protein, with an exposed Myr, was equilibrated separately for 150 ns to allow the sequestration of Myr in the internal hydrophobic cavity of the protein. Equilibrated coordinates of each bilayer and the protein were merged with the protein at least 1.0 nm above the bilayer using GROMACS. The protein-membrane systems started with a sequestered Myr MA and were simulated for 200 ns on GROMACS. Coordinates from equilibrated systems with the *inner* and *raft* models were extended to microsecond-trajectories on Anton2 to characterize Myr exposure and insertion events. Additionally, these bilayers were simulated starting from an inserted Myr configuration to examine its long-term effect on membrane dynamics. The *control* model was also simulated on Anton2 starting from an inserted Myr configuration to compare the effects of the lipid tail on membrane properties when cholesterol is not present.

The equilibrated membranes were later replicated to form larger membrane patches to simulate three MA monomers or a formed MA trimer near the membrane surface. The smaller membrane patches contained 150 lipids per leaflet, while the larger patches have 600 lipids per leaflet with a

surface of 282.24 nm$^2$ (16.8 x 16.8 nm). The MA trimer (PDBID:1HIW) corresponding to the structure published by *Hill et al., 1996* was equilibrated in water separately and then introduced near the membrane surface. During the trimer-only equilibration, two Myr tails in individual units were sequestered into the hydrophobic cavity and one remained outside. This equilibrated structure was positioned 1 nm away from the *inner* and *raft* large bilayers and simulated for 1 $\mu s$. Lastly, three separate monomers with exposed Myr were positioned near the bilayer and within 2.5 nm of each other to examine their interaction with other protein units as well as with the bilayer without any bias. Overall, we simulated 13 systems for a total of 42.8 μs, including the short 200 ns trajectories of an MA monomer on the small membrane patches.

Simulations were carried using isobaric and isothermal dynamics (constant NPT ensemble) with a simulation timestep of 2 fs and periodic boundary conditions. The temperature was kept constant at 310.15 K using the Nose-Hoover thermostat (*Hoover, 1985*; *Nosé, 1984*) with a coupling time constant of 1.0 ps in GROMACS. Similarly, the pressure was set at 1 bar and controlled using the Parrinello-Rahman barostat semi-isotropically with a compressibility of $4.5 \times 10^{-5}$ and a coupling time constant of 5.0 ps (*Nosé and Klein, 1983*; *Parrinello and Rahman, 1981*). Van der Waals interactions were computed using a switching function between 1.0 and 1.2 nm, and long-range electrostatics evaluated using Particle Mesh Ewald (*Darden et al., 1993*). Hydrogen bonds in GROMACS were constrained using the LINCS algorithm (*Hess et al., 1997*). Simulation parameters for the Anton2 machine are set by its ark guesser files (scripts designed to optimize the parameters for the integration algorithms to enhance numerical integration during the simulation); as such, the cut-off values to compute interactions between neighboring atoms are selected automatically during system preparation. Long-range electrostatics were computed using the Gaussian Split Ewald algorithm (*Shan et al., 2005*), and hydrogen bonds constrained using the SHAKE algorithm (*Ryckaert et al., 1977*). Finally, the Nose-Hoover thermostat and MTK barostat control the temperature and pressure respectively during NPT dynamics using optimized parameters set by the *Multigrator* integrator on Anton2 (*Lippert et al., 2013*).

Analysis of the trajectories was performed using GROMACS, MDAnalysis (*Gowers et al., 2016*; *Michaud-Agrawal et al., 2011*), MDTraj (*McGibbon et al., 2015*), and PyEmma (*Scherer et al., 2015*) packages. Visual Molecular Dynamics (VMD) software package (*Humphrey et al., 1996*) was used to visualize the systems and generate all the snapshots. Description of the analysis included in the next sections is summarized on Table S3 in *Supplementary file 1*. All reported values are blocked averages with their respective standard error for each given quantity over at least 100 ns of equilibrated trajectory unless stated otherwise. In general, distances were computed between the center-of-mass of the corresponding atom(s); angles were computed defining vectors between the center-of-mass of the corresponding initial and final residues; the bilayer center was estimated as the center-of-mass between the phosphorus atoms in each leaflet leaflets. Time series plots show the raw data faded in the background and the moving average, computed every 30 data points, as bolded lines for clarity. Lastly, we carried time-lagged independent component analysis (tICA) (*Pérez-Hernández and Noé, 2016*; *Pérez-Hernández et al., 2013*; *Sultan and Pande, 2017*) as a reduction technique to quantify the protein changes during Myr insertion. We selected the two slowest independent components (tICs) computed as the linear combination of the distance between alpha carbons in the protein and the carbon atoms of the Myr tail. As an estimate of the relative contributions of the tICs to the Myr conformation or state, that is sequestered, exposed, inserted, we computed the natural log of the probability densities of the projected trajectories onto the corresponding tIC.

## Conclusions

We examined specific interactions between the MA domain of HIV-1 Gag polyprotein and lipid bilayers using all-atom molecular dynamics simulations. Our conclusions provide insight on the effect of bilayer nature on MA-binding events as well as a mechanism for Myr insertion. MA binding is largely stabilized by electrostatic interactions between the HBR and charged lipid headgroups, specially PIP$_2$ lipids. Binding is permanent and takes places early in the simulation with no need of Myr insertion as an 'anchor.' From the two bound conformation observed in our systems, only the open conformation is feasible if the rest of the Gag polyprotein were present; it is only from this conformation that Myr insertion takes place.

Our simulations are the first of their kind to capture repeatedly Myr insertion through unbiased all-atom MD simulations. Through our analysis, we show the time scale for insertion is on average 50 ns, and only occurs when the protein is bound in what we characterize as the open conformation: a flat HBR on the membrane, H5 extended towards the solvent, and the mouth of the protein free for Myr release form the hydrophobic cavity. As Myr unrolls into the binding leaflet, the first helix in the protein sequence, H1, shifts to rest flat on the membrane surface. The first and last carbons in the Myr tail, C2 and C14 respectively, pass each other as the tail inserts into the leaflet. Depending on the location of Myr prior to insertion, C14 can extend vertically toward the solvent or enter the leaflet 'sweeping' the membrane surface as Myr unrolls into the leaflet from a flat position between the protein and the membrane. This mechanism was further characterized using tICA, identifying the HBR, H1, and H5 as key protein regions that modulate or contribute to the insertion process. Importantly, we built three membrane models to study the effect of different membrane mechanical and structural properties on insertion events. We found Myr only inserts into our *inner* membrane model, that mimics the inner leaflet of the PM in terms of anionic lipids, lipid tail unsaturation, and sterol content. From these studies, we conclude that a combination of membrane surface charge, lipid packing, and the degree of lipid tail unsaturation inside the bilayer are key determinants for Myr insertion. From *Table 1*, the surface charge for the *inner* and *control* models are similar, yet the presence of sterol in the *inner* model results in an overall more saturated and ordered environment that Myr prefers. Packing defects on the membrane surface also play a role; one expects more defects on the surface of the *control* model due to its lack of cholesterol and more fluid nature, but the unsaturation degree and disorder of the membrane hydrophobic core do not seem to provide an optimum or attractive place for Myr. Finally, the *raft* model with high sterol and sphingomyelin content seems too tightly packed and ordered for any insertion event to occur, even with multiple MA units.

The protein was randomly positioned near the membrane surface after a short equilibration in solvent, so we have not quantified the effect of lipid domains on initial protein binding or Myr insertion. Such a study would provide additional insights on the preference of MA for lipid order or disordered domains. We have shown, however, that insertion of Myr results in modifications of lipid order as quantified by the order parameter $S_{CD}$. The effect is not as dramatic for the case of a single monomer, which alters the order of the last carbons in the long sphingomyelin lipid tails. However, the presence of a formed trimer that enables quicker Myr insertion, induces increase in order in the mid-region of the same lipids. Our results present evidence that an increasing number of protein units enhances the changes observed on the membrane upon MA binding, as would be the case at early stages of the viral assembly process of HIV-1 in the cell. Particularly, protein-lipid interactions recruit specific lipid species, like $PIP_2$, to the protein binding site changing the local lipid composition. In turn, this also changes the local membrane properties and can influence the recruitment of other crucial components for the virus, creating a more suitable platform to continue and complete viral assembly and posterior budding. For example, recruitment of additional Gag units and other viral proteins that are known to partition to specific lipid environments (saturated/unsaturated lipid tails).

One of the main characteristics of the PM is the asymmetry between its bilayers; here, we examined symmetric membrane models and reserved a study with asymmetric models for a separate publication. The study with an asymmetric model aims to examine the modifications to lipid-lipid interactions between leaflets due to binding of multiple MA units in an effort to further understand the dynamics of early viral assembly. Conclusions from the present work serve to reveal the specific MA-lipid contacts, effects of membrane nature on protein binding and Myr insertion events, and membrane response to protein binding. The MA domain is a conserved domain of Gag across several retroviruses, thus we expect our conclusions to also provide insight into the mechanism of binding in other viral systems (*Mattei et al., 2018*). Fundamental understanding of the membrane targeting mechanism of MA can offer new perspectives for inhibitor-based antiretroviral treatments (*Zentner et al., 2013*). Furthermore, several studies look at how the acyl tail modulates protein-membrane interactions, from driving the interaction to a specific cellular compartment to modifying the local membrane environment (*Brunsveld et al., 2009*; *Gohlke et al., 2010*; *Thukral et al., 2015*; *Ray et al., 2017*). This work contributes to the study of lipidated peripheral proteins and their interactions with membranes in the context of macromolecular assembly.

## Acknowledgements

This work was funded by the National Institute of General Medical Sciences of the United States NIH under grant R01GM063796. The authors thank Drs. A J Pak, F Aydin, S Kim, and A Yu for consultation regarding the tICA methodology and general discussions on HIV-1 dynamics throughout the duration of the study and manuscript preparation. Computational resources were provided by the Pittsburgh Super Computing Center through the Anton two machine under Grant R01GM116961 from the National Institutes of Health, and the specific allocation PSCA17046P. The Anton two machine at PSC was generously made available by DE Shaw Research. Part of this work was also completed with resources from the University of Chicago Research Computing Center, and the Extreme Science and Engineering Discovery Environment, supported by the National Science Foundation grant number ACI-1548562.

## Additional information

### Funding

| Funder | Grant reference number | Author |
| --- | --- | --- |
| National Institute of General Medical Sciences | R01GM063796 | Viviana Monje-Galvan<br>Gregory A Voth |
| National Institutes of Health | R01GM116961 | Gregory A Voth |
| National Science Foundation | ACI-1548562 | Gregory A Voth |

The funders had no role in study design, data collection and interpretation, or the decision to submit the work for publication.

### Author contributions

Viviana Monje-Galvan, Formal analysis, Investigation, Writing - original draft, Writing - review and editing; Gregory A Voth, Conceptualization, Supervision, Funding acquisition, Investigation, Visualization, Writing - review and editing

### Author ORCIDs

Viviana Monje-Galvan (iD) https://orcid.org/0000-0002-9202-782X
Gregory A Voth (iD) https://orcid.org/0000-0002-3267-6748

### Decision letter and Author response

Decision letter https://doi.org/10.7554/eLife.58621.sa1
Author response https://doi.org/10.7554/eLife.58621.sa2

## Additional files

### Supplementary files

• Supplementary file 1. Tables S1-S3 cited and discussed in this work. These list detailed information about all the systems of study, and thorough description of the quantitative analysis performed in the collected simulation data.

• Transparent reporting form

### Data availability

The simulation trajectories used for the analysis presented in this work are available at the Pittsburgh Supercomputing Center (PSC) Database for simulations run on the Anton2 Machine (http://psc.edu/anton-project-summaries?id=3071&pid=34).

The following dataset was generated:

| Author(s) | Year | Dataset title | Dataset URL | Database and Identifier |
|---|---|---|---|---|
| Monje-Galvan V, Voth GA | 2019 | All-atom molecular dynamics simulations of the matrix domain of HIV-1 Gag protein and model membranes | http://psc.edu/anton-project-summaries?id=3071&pid=34 | Pittsburgh Supercomputing Center Public Repository, 34 |

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
