## [Decision Letter]

**Acceptance summary:**

The reviewers and editors recognized that this study is both important and timely. The membrane interaction and self-assembly of the Gag protein is a critical step in viral capsid assembly. This is an area that has received considerable attention over the past three decades, yet much has remained unknown due to the complexity of this interaction. The insights derived from this systematic analysis based on molecular simulation data constitute a significant advance and are likely to foster novel research in this area.

**Decision letter after peer review:**

Thank you for submitting your article "Binding mechanism of the matrix domain of HIV-1 gag to lipid membranes" for consideration by *eLife*. Your article has been reviewed by three peer reviewers, whose comments are provided below. The evaluation has been overseen by José Faraldo-Gómez as the Senior and Reviewing Editor. The reviewers have opted to remain anonymous.

The reviewers have discussed the reviews with one another and with the Senior Editor. Based on this discussion, we would like to invite you to submit a revised version of your manuscript that convincingly addresses the concerns and questions raised. Please note the reviewers have asked to re-evaluate your revised manuscript before making a final recommendation on whether or not it should be published in *eLife*. I therefore urge you to give careful consideration to the revisions of the manuscript as well as your response.

Reviewer #1:

This is an important and timely study that describes novel all-atom MD simulations of interactions and structural changes associated with membrane binding by the MA domain of the HIV-1 Gag protein. This is an area that has received considerable experimental, and some computational, attention over the past three decades, yet much remains unknown due to the complexity of the interactions. Voth and co-workers have done a beautiful and systematic job of parameterizing plasma membrane interactions with MA and then examining the influence of the membrane constituents (Phosphatidylinositol-4,5-bisphosphate, phosphatidyl serine, cholesterol, and others) on membrane binding. They conducted MD trajectories at timescales (40 us) sufficient to follow insertion of the N-terminal myristoyl group of MA into the membrane, and characterize the influence of up to six membrane constituents known to be important for targeting MA specifically to the plasma membrane. Interestingly, all simulations were initiated with the protein in the "myristate sequestered" conformation, and stable sequestration of the myristate chain within the membrane only occurred for membranes homologous to the plasma membrane. Their studies also reveal how MA-MA interactions, known to be important for recruitment of the viral envelope protein, promote reorganization of inner-leaflet membrane constituents.

This appears to me to be a comprehensive and well-executed study. It should be of broad interest to the biomedical community, particularly to groups interested in HIV and protein-membrane interactions. My only recommendation is that the Discussion section include some speculation on the recently discovery by Bieniasz and co-workers that, in the cytoplasm, the MA domain is bound to tRNAs (predominantly tRNALys), and by Summers and co-workers that tRNA binding is tight enough to inhibit binding to liposomes. I'm certainly not suggesting that additional experiments with tRNAs need to be conducted for this paper. But it would be good for the readers to know that, although these studies provide fundamental new insights into the changes and interactions that must happen during virus assembly, the actual mechanism in cells is probably more complex.

Also, it would be really nice if the authors could provide a supplementary video showing one of their trajectories showing MA docking with the membrane (the biologically relevant conformation) and the myristate transitioning from the protein pocket to the membrane.

Reviewer #2:

This paper reports extensive molecular dynamics simulations of the Gag protein interacting with three different membranes: a model of the inner plasma membrane (PM), a cholesterol-free version of the PM, and a lipid raft. It is shown convincingly using unbiased simulations that the myristoylated (Myr) N-terminus of the proteins selectively binds to the PM model. This result and accompanying analysis provides valuable insights into the interactions driving membrane binding, and highlights the importance of selecting appropriate membrane models for such studies. I recommend publication following minor revisions.

I read this paper several times looking for a concise summary-like statement of why Myr preferentially inserts in the inner PM model, but could not find one. Mention is made of PIP2, but the PIP2 concentration is lowest in this model. Cholesterol in critical, but this is included in the raft model. Charge is important, but is it specific are nonspecific interactions of the protein and DOPS? From Supplementary Table 1 the density of the inner PM and raft models appears to be comparable. Is it the replacement of POPE for BSM? Sort this out and write it clearly in the conclusions and the Abstract, don't make readers hunt for it.

Move the Supplementary Table 1 (compositions of the three models) to the main text. It was very hard to follow the arguments in the paper without this table handy. (If there is a space issue, the main text is somewhat wordy and should be easy to trim.) Some other questions/issues with this table:

1) The columns labelled "small" and "large" contain the same numbers of water and ions. Is this a mistake?

2) Include the charge density for each system. What charge is being used for PIP2? Is the + sign in the PIP2 for the chol-free membrane a typo?

3) Let the reader know that the B in BSM stands for behenic acid. (I initially thought it was a misprint of the more common PSM)

4) Perhaps incorrectly I believed that PIP2 is not a component of lipid rafts, or at least the liquid ordered phase. A sentence on the raft composition would helpful.

It would be useful to note studies showing that acyl chains readily insert into numerous membranes. Acyl chains of different RAS insert into liquid ordered phases, liquid disordered phases and the boundaries between them. Hence the interaction of Gag and the PM is special.

Reviewer #3:

In their manuscript "Binding mechanism of the matrix domain of HIV-1 gag to lipid membranes" Monje-Galvan and Voth present extensive simulations (totaling some 40 uses) of the MA domain of the HIV Gag protein interacting with several different model membranes. The membrane interaction and self-assembly of Gag at the inner leaflet is a critical step in viral capsid assembly, which provides a major motivation for the work. The authors compare interaction of the MA domain and insertion (or non-insertion) of its myristoyl anchor in simulations of different model membranes, designed to mimic different subcellular localizations (inner leaflet, raft-like, etc), which is a strength of the work. However, although the technical aspects of the work are sound, the broader implications in terms of mechanism are not clear.

1) What critical new information has been gained regarding capsid assembly, and how does it bear on existing models of assembly? Although there is a good deal of highly technical information in the manuscript, it is never synthesized into a "big picture" for capsid assembly. It is therefore unlikely to find a broad audience, and instead be of interest mainly to the membrane and membrane protein simulation community. What new simulations and experiments do the authors' results and analysis motivate?

2) Although the use of distinct lipid mixtures to compare MA interaction with different membrane regions is overall a strength, the choice of lipid mixtures is not well motivated or explained. There are hardly any lipids like DOPC or DOPS (ie, with identical tails at the sn-1 and sn-2 positions) in mammalian lipidomes. Also, what is the "raft" membrane model supposed to mimic? An inner leaflet raft? If so, on what basis was this particular composition chosen?

3) There is some discussion of Myr insertion as a mechanism to initiate sphingolipid clustering, but this is neither quantified nor discussed in the context of mechanism or function.

---

## [Author Response]

Reviewer #1:[…] This appears to me to be a comprehensive and well-executed study. It should be of broad interest to the biomedical community, particularly to groups interested in HIV and protein-membrane interactions. My only recommendation is that the Discussion section include some speculation on the recently discovery by Bieniasz and co-workers that, in the cytoplasm, the MA domain is bound to tRNAs (predominantly tRNALys), and by Summers and co-workers that tRNA binding is tight enough to inhibit binding to liposomes. I'm certainly not suggesting that additional experiments with tRNAs need to be conducted for this paper. But it would be good for the readers to know that, although these studies provide fundamental new insights into the changes and interactions that must happen during virus assembly, the actual mechanism in cells is probably more complex.Also, it would be really nice if the authors could provide a supplementary video showing one of their trajectories showing MA docking with the membrane (the biologically relevant conformation) and the myristate transitioning from the protein pocket to the membrane.

We thank the reviewer for the positive feedback and welcome the suggestion to include a supplementary video that shows MA binding and Myr insertion into one of our bilayer models. We generated two short clips of a sample system showing (i) the initial MA binding, and the (ii) the permanent insertion of Myr. Reference to these videos was added to the main text as shown below, and a title and legend provided at the end of the article as indicated in the *eLife* Author Guide.

“The initial conformation of the protein did not influence the final bound state. Video 1 shows an example where MA bound in the open conformation, the HBR loop interacts with the membrane and stablishes a permanent bound state from which Myr can exit the protein’s hydrophobic cavity.”

“Video 2 shows a portion of the trajectory in which Myr is outside the hydrophobic cavity of the protein, interacts with the solvent and lipid headgroups, and then inserts into the bilayer.”

Information added at the end of the article:

**“**Video 1 – “MA binding in the open conformation.” […] The tail can survey the membrane surface and even extend towards the water before permanently inserting into the bilayer.”

We also corrected the line in the original manuscript, as pointed out by the reviewer, to correctly read: ‘…and release of an immature virion enclosing the viral genome…’ instead of ‘viral genetic code.’

Reviewer #2:[…] I read this paper several times looking for a concise summary-like statement of why Myr preferentially inserts in the inner PM model, but could not find one. Mention is made of PIP2, but the PIP2 concentration is lowest in this model. Cholesterol in critical, but this is included in the raft model. Charge is important, but is it specific are nonspecific interactions of the protein and DOPS? From Supplementary Table 1 the density of the inner PM and raft models appears to be comparable. Is it the replacement of POPE for BSM? Sort this out and write it clearly in the conclusions and the Abstract, don't make readers hunt for it.

We thank the reviewer for these comments and insightful questions. We added to the Abstract and Conclusions sections as shown below. We would like to point out that both the *inner* and the *raft* models were designed to mimic mechanical and structural properties of the PM. As such, both are high in sterol content (nearly 30 mol%) but have charge and other differences. Namely, the *inner* model has 25 mol% of PE lipids, known to reside mainly in the inner leaflet of the PM, and nearly three times the amount of DOPS lipids than the *raft* model. On the other hand, the *raft* model, built to reflect the existence of lipid rafts (or regions of high order) in the PM, lacks PE lipids and is rich in sphingomyelin and cholesterol (nearly 60 mol%), both the main components of rafts. In addition, these models were built after the experimental studies by Yandrapalli et al., 2016, to allow us a more direct comparison as we cite and discuss in the main manuscript. They observed Gag binding regions with enriched sphingomyelin, cholesterol, and PIP_2_; according to more recent studies of our experimental collaborator (unpublished), PIP_2_ would co-localize to the protein binding site in the inner leaflet of the PM, and modulate the behavior of lipids on the outer leaflet by ‘recruiting’ or aligning with regions of lipid rafts in the outer membrane. This mechanism is currently under study and we are analyzing simulations with an asymmetric model for the PM, which we plan to publish with our experimental collaborators.

“We characterized Myr insertion events from microsecond trajectories, and examined the membrane response upon initial membrane targeting by MA. Insertion events only occur with one of the membrane models, showing a combination of surface charge and internal membrane structure modulate this process.”

“We found Myr only inserts into our *inner* membrane model, that mimics the inner leaflet of the PM in terms of anionic lipid, lipid tail unsaturation, and sterol content. […] Finally, the *raft* model with high sterol and sphingomyelin content seems too tightly packed and ordered for any insertion event to occur, even with multiple MA units.”

Move the Supplementary Table 1 (compositions of the three models) to the main text. It was very hard to follow the arguments in the paper without this table handy. (If there is a space issue, the main text is somewhat wordy and should be easy to trim.) Some other questions/issues with this table:

Supplementary Table 1 was moved to the main manuscript (Table 1) changes to the manuscript are listed under each point mentioned by the reviewer.

Table 1: The table has been moved to the main manuscript and all the cross-references adjusted.

1) The columns labelled "small" and "large" contain the same numbers of water and ions. Is this a mistake?

Table 1: Yes, this was a typo and now the table lists the correct values for the large systems.

2) Include the charge density for each system. What charge is being used for PIP2? Is the + sign in the PIP2 for the chol-free membrane a typo?

Table 1: A column for the charge density per membrane surface has been added to the table. The plus sign next to *PIP_2_* was reference to a footnote listing the actual lipid species of this model (SAPI25, 18:0 – 20:4, as per the IUPAC numerical symbols for fatty acid tail (un)saturation), and its chemical structure shown in Figure 2—figure supplement 1. This has been updated to avoid confusion.

3) Let the reader know that the B in BSM stands for behenic acid. (I initially thought it was a misprint of the more common PSM)

Table 1: We have included a legend at the foot of the table listing the full names of all the lipid species used in our models. The same names have been listed at the bottom of Figure 2—figure supplement 1, which shows all the corresponding chemical structures. “BSM” stands for brain sphingomyelin, the full name for the particular species in our model is N-octadecanoyl-D-erythro-sphingosylphosphorylcholine. *Behenic* acid has a 22-carbon backbone, whereas the fatty acid tails in *BSM* are 16 and 18, respectively.

4) Perhaps incorrectly I believed that PIP2 is not a component of lipid rafts, or at least the liquid ordered phase. A sentence on the raft composition would helpful.

The reviewer is correct. PIP_2_ is not a component of lipid rafts; it prefers the liquid disordered phase due to its polyunsaturated tail. In the context of these comments, we attribute the reviewer’s question to the inclusion of this species in the *raft* model. This model is built to reflect the ability of the PM lipids to form rafts; to this end the model contains nearly 60 mol% of lipids that are widely known to aggregate into rafts, i.e., sphingomyelin and cholesterol. DOPC is included as “bulk” lipid, conforming another 30% of the lipids in this model. DOPS and PIP_2_ are included to account for the overall anionic lipid content in the PM. We added the following sentence to the first paragraph in the Materials and methods section about the lipid composition of our model membranes:

“The *inner* model mimics the anionic character of the PM, while the *raft* model emphasizes the ability of the PM to phase separate and form lipid clusters enriched in sphingomyelin and cholesterol lipids. […] On the other hand, the *raft* model, built to reflect the existence of lipid rafts (or regions of high order) in the PM, lacks PE lipids but is rich in sphingomyelin and cholesterol (nearly 60 mol%), both the main components of rafts. PIP_2_ and DOPS lipids are included to account for the anionic content of the membrane.”

It would be useful to note studies showing that acyl chains readily insert into numerous membranes. Acyl chains of different RAS insert into liquid ordered phases, liquid disordered phases and the boundaries between them. Hence the interaction of Gag and the PM is special.

We added the following comment to the “Conclusions” subsection with corresponding references to highlight the conclusions from our work are applicable to similar systems relevant in cell signaling and other processes:

“The MA domain is a conserved domain of Gag across several retroviruses, thus we expect our conclusions to also provide insight into the mechanism of binding in other viral systems (1). […] This work contributes to the study of lipidated peripheral proteins and their interactions with membranes in the context of macromolecular assembly.”

Reviewer #3:In their manuscript "Binding mechanism of the matrix domain of HIV-1 gag to lipid membranes" Monje-Galvan and Voth present extensive simulations (totaling some 40 uses) of the MA domain of the HIV Gag protein interacting with several different model membranes. The membrane interaction and self-assembly of Gag at the inner leaflet is a critical step in viral capsid assembly, which provides a major motivation for the work. The authors compare interaction of the MA domain and insertion (or non-insertion) of its myristoyl anchor in simulations of different model membranes, designed to mimic different subcellular localizations (inner leaflet, raft-like, etc), which is a strength of the work. However, although the technical aspects of the work are sound, the broader implications in terms of mechanism are not clear.1) What critical new information has been gained regarding capsid assembly, and how does it bear on existing models of assembly? Although there is a good deal of highly technical information in the manuscript, it is never synthesized into a "big picture" for capsid assembly. It is therefore unlikely to find a broad audience, and instead be of interest mainly to the membrane and membrane protein simulation community. What new simulations and experiments do the authors' results and analysis motivate?

This study was geared to understanding the mechanisms of Gag membrane targeting via binding of its MA domain to the PM. Our studies are setup during very early stages of viral assembly, i.e., a single MA unit or a maximum of three units on the membrane surface. Capsid assembly occurs during maturation, after viral assembly and budding; in the context of viral assembly on the PM, we can see statistically relevant differences in membrane structure when MA proteins are on the membrane surface, see Figure 6 for a quantitative measure of lipid order parameters, S_CD_. We are currently running simulations with more MA units and an asymmetric model for the PM in collaboration with an experimental group to quantify changes during early viral assembly. We aim to show aggregation of MA units, Gag proteins in the physical system, modifies the local membrane environment and results in a new distribution of lipids around the proteins, which in turn recruit crucial components for the virus in preparation for budding. We added the following sentence to the “Conclusions” subsection:

“… as would be the case at early stages of the viral assembly process of HIV-1 in the cell. Specifically, protein-lipid interactions that recruit specific lipid species to the protein binding site result in a change in local lipid composition. In turn, this also changes the local membrane properties and further recruits other crucial components for the virus to create a more suitable platform to continue and complete viral assembly and posterior budding.”

“One of the main characteristics of the PM is the asymmetry between its bilayers; here, we examined symmetric membrane models and reserved a study with asymmetric models for a separate publication. The study with an asymmetric model aims to examine the modifications to lipid-lipid interactions between leaflets due to binding of multiple MA units in an effort to further understand the dynamics of early viral assembly.”

2) Although the use of distinct lipid mixtures to compare MA interaction with different membrane regions is overall a strength, the choice of lipid mixtures is not well motivated or explained. There are hardly any lipids like DOPC or DOPS (i.e., with identical tails at the sn-1 and sn-2 positions) in mammalian lipidomes. Also, what is the "raft" membrane model supposed to mimic? An inner leaflet raft? If so, on what basis was this particular composition chosen?

As mentioned in the beginning of the Results section, we built the membrane models after an experimental study from Yandrapalli et al., 2016. We added a few sentences to the Materials and methods section highlighting the differences and choice of lipids for the *inner* and *raft* models, also discussed in the response to Reviewer 2. Both the *inner* and the *raft* models were designed to mimic mechanical and structural properties of the PM, and have nearly 30% of sterol content. The *inner* model has 25 mol% of PE lipids, known to reside mainly in the inner leaflet of the PM, and nearly three times the amount of DOPS lipids than the *raft* model. The *raft* model, built to reflect the existence of lipid rafts (or regions of high order) in the PM, lacks PE lipids and is rich in sphingomyelin and cholesterol (nearly 60 mol%), both the main components of rafts.

3) There is some discussion of Myr insertion as a mechanism to initiate sphingolipid clustering, but this is neither quantified nor discussed in the context of mechanism or function.

We are currently working with an experimental collaborator examining sphingolipid recruitment and clustering on the outer leaflet of the PM and how this may be initiated or regulated by PIP_2_ recruitment and binding to MA (Gag) binding sites using an asymmetric model for the PM simulations. Lipid clustering was out of the scope of this paper, and we have updated our word choice and changed *“clustering”* to *“enriched regions”* to prevent confusion.

Reference:

1) Mattei, S., A. Tan, B. Glass, B. Müller, H.-G. Kräusslich, and J. A. G. Briggs. 2018. High-resolution structures of HIV-1 Gag cleavage mutants determine structural switch for virus maturation. Proceedings of the National Academy of Sciences 115(40):E9401.